# CONTINUOUS SPIKING GRAPH ODE NETWORKS

## ABSTRACT

Spiking Graph Networks (SGNs), as bio-inspired neural models that address energy consumption challenges for graph classification, have attracted considerable attention from researchers and the industry. However, SGNs are typically applied in static scenarios with real-valued inputs and cannot be directly utilized for dynamic prediction because of their limited capacity to handle dynamic real-valued features, denoted as architectural inapplicability. Moreover, they suffer from accuracy loss due to the inherently discrete nature of spike-based representations. Inspired by recent graph ordinary differential equation (ODE) methods, we propose the framework named **C**ontinuous **S**piking **G**raph **O**DE Networks (CSGO), which leverages the advantages of graph ODE to address the architectural inapplicability, and employs high-order structures to solve the problem of information loss. Specifically, CSGO replaces the high energy-consuming static SGNs with an efficient Graph ODE process by incorporating SGNs with graph ODE into a unified framework, thereby achieving energy efficiency. Then, we derive a high-order spike representation capable of preserving more information. By integrating this with a high-order graph ODE, we propose the second-order CSGO to address the information loss challenge. Furthermore, we prove that the second-order CSGO maintains stability during the dynamic graph learning. Extensive experiments validate the superiority of the proposed CSGO in performance while maintaining low power consumption.

## 1 INTRODUCTION

Spiking Graph Networks (SGNs) (Zhu et al., 2022; Xu et al., 2021) are a type of artificial neural network specifically designed to process graph information in a manner similar to the human brain. SGNs typically transform static and real-valued graph features into discrete spikes, and then emulate the neuron's charging and discharging processes to achieve spikes representation for graph classification. The distinctive features of SGNs is their ability to capture semantic spiking representations while maintaining low energy consumption, making them well-suited for event-based processing tasks (Yao et al., 2021), such as object recognition (Gu et al., 2020; Li et al., 2021), real-time data analysis (Zhu et al., 2020b; Bauer et al., 2019), and graph classification (Li et al., 2023; Zhu et al., 2022; Xu et al., 2021).

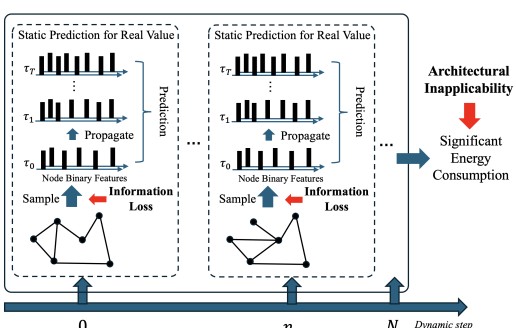

Figure 1: Illustration of information loss and architecture inapplicability. Information loss arises from the sampling of real-value inputs, while the architecture inapplicability refers to the unsuitability of dynamic real-valued SGN methods.

SGNs are commonly applied in scenarios that handle static real-valued or continuous binary inputs (Guo et al., 2022; Wang et al., 2022; Lv et al., 2023). For real-valued features, SGNs initially sample binary features on each node using a Bernoulli Distribution (Zenke & Ganguli, 2018), where the probability corresponds to the feature's real values. These features are then propagated by simulating the neuron's charging and discharging processes. However, limited research has explored the application of SGNs to dynamic real-valued inputs. Typically, simply sampling binary features and calculating the spiking representation at each dynamic step results in significant energy consumption, rendering it unsuitable for low-power devices. Additionally, the inherent characteristic of SGNs that

transforms continuous features into binary representations leads to information loss and performance degradation (Yan et al., 2021). As shown in Figure 1, the *architectural inapplicability* and *information loss* problems limit the application of SGNs for dynamic graph prediction in real-world scenarios.

Inspired by recent developments in graph ordinary differential equations (Graph ODEs) (Battaglia et al., 2016; Kipf et al., 2018), which typically use GNNs to obtain node representations through ordinary differential equations, and recognizing that higher-order neural networks can preserve more information (Luo et al., 2023), we propose utilizing high-order SGNs and Graph ODEs to address these challenges. However, incorporating Graph ODE with SGNs for dynamic prediction tasks is difficult due to the following challenges: (1) *How to efficiently incorporate the SGNs and Graph ODE into a unified framework?* SGNs and Graph ODEs process information along two distinct dimensions: the latency dimension in SGNs and the dynamic evolution in graph-based models. A key challenge is how to merge these dual processes into a cohesive framework that maintains the energy-efficient properties of SGNs while leveraging the dynamic learning capabilities of Graph ODEs. (2) *How to alleviate the problem of information loss of SGNs?* SGNs achieve low power consumption by discretizing continuous features, but this binarization often leads to a significant loss of fine-grained information, reducing performance. Addressing the information loss in SGNs is therefore another critical challenge. (3) *How to guarantee the stability of the proposed framework?* Traditional graph-based techniques encounter the challenge of exploding and vanishing gradient problems when modeling the dynamic evolution of GNNs. Therefore, devising a stable model for graph learning with theoretical guarantee constitutes the third prominent challenge.

To tackle these challenges, we propose a framework named **C**ontinuous **S**piking **G**raph **O**DE Networks (CSGO) for dynamic graph learning tasks. Specifically, to address the first challenge, we approach it by considering SGNs as a type of ODE and integrating it with Graph ODE into a framework, denoted as CSGO-1st. The CSGO-1st structure models the initial representation using SGNs by considering the inner time latency at each dynamic step and evaluates the dynamics evolution with Graph ODE iteratively. Furthermore, to address the information loss problem caused by SGNs, we derive a high-order spike representation with second-order SGNs structure and incorporate with high-order Graph ODE, which referred to as CSGO-2nd. Moreover, we provide the theoretical guarantee that CSGO successfully mitigates issues related to exploding and vanishing gradients. We perform comprehensive evaluations of CSGO against state-of-the-art methods on various graph-based datasets, showcasing the efficacy and versatility of the proposed approach.

In summary, the contributions can be summarized as follows: (1) **Problem Setup.** We present a novel problem in SGNs, emphasizing the challenge of achieving high performance while maintaining low power consumption for dynamic graph classification tasks. (2) **Novel Architecture**. We propose the CSGO, which efficiently incorporates SGNs and Graph ODE into a unified framework, retaining the energy-efficient properties of SGNs while preserving the ability to capture dynamic changes in Graph ODE. Furthermore, we first derive the second-order spike representation and study the backpropagation of second-order SGNs to mitigate the information loss problem. (3) **Theoretical Analysis**. We provide a theoretical proof demonstrating that CSGO effectively mitigates the issue of exploding and vanishing gradients, ensuring the stability of our proposed method. (4) **Extensive Experiments**. We evaluate the proposed CSGO on extensive graph-based learning datasets, which evaluate that our proposed CSGO outperforms the variety of state-of-the-art methods.

## 2 RELATED WORK

**Graph Ordinary Differential Equations.** Recently, numerous methods based on dynamic GNNs have emerged for modeling dynamic interaction systems (Battaglia et al., 2016; Kipf et al., 2018; Chen et al., 2018; Ju et al., 2024). These methods commonly employ GNNs to initialize node representations at discrete timestamps, which are then utilized for predicting node behaviors. Nevertheless, these discrete methodologies often necessitate the presence of all nodes at each timestamp (Huang et al., 2020; 2021; Yin et al., 2023a; 2022), which is challenging to achieve in real-world scenarios. In contrast, ODE has proven to be effective in modeling system dynamics when dealing with missing data (Chen et al., 2018). Recent works (Poli et al., 2019; Gupta et al., 2022) involve initializing state representations with GNNs, followed by the establishment of a neural ODE model for both nodes and edges, guiding the evolution of the dynamical system. Additionally, high-order correlations (Luo et al., 2023; Zhang et al., 2022) have been shown to efficiently model the dynamic evolution of graphs.

We integrate energy-efficient SNNs into Graph ODE, thereby retaining the low energy characteristics of SNNs while harnessing the dynamic learning capabilities of Graph ODE.

**Spiking Graph Networks.** SGNs (Zhu et al., 2022; Xu et al., 2021) have emerged as a promising solution for addressing energy consumption challenges on graph classification tasks. Recently, various SGNs (Xu et al., 2021; Wang & Jiang, 2022; Zhu et al., 2022) have demonstrated low energy consumption and high bio-fidelity. These models employ similarly reactive spiking neurons (Gerstner & Kistler, 2002) to process propagated graph data, achieving both low energy consumption and maintained bio-fidelity. However, these methods are generally applied in the scenarios of static real-valued or continuous binary inputs (Guo et al., 2022; Wang et al., 2022; Lv et al., 2023). There is still limited research focusing on dynamic spiking graphs with real-valued inputs. Although some works have attempted to apply static SNNs to dynamic graphs (Li et al., 2023; Yin et al., 2024) by calculating the static spiking representation at each dynamic step and then predicting the evolution with a new spiking layer, these methods generally demand substantial energy consumption, rendering them impractical for low-power devices. Our approach ingeniously combines SGNs with Graph ODE to effectively capture the dynamic changes while maintaining low power consumption.

# 3 PRELIMINARIES

## 3.1 DYNAMIC GRAPH NEURAL NETWORKS

**Problem Formulation:** Given a graph $G = (\mathcal{V}, \mathcal{E})$ with the node set $\mathcal{V}$ and the edge set $\mathcal{E}$. $\mathbf{X} \in \mathbb{R}^{|\mathcal{V}| \times d}$ is the node feature matrix, $d$ is the feature dimension. The binary adjacency matrix denoted as $\mathbf{A} \in \mathbb{R}^{|\mathcal{V}| \times |\mathcal{V}|}$, where $a_{ij} = 1$ denotes there exists a connection between node $i$ and $j$, and vice versa. Our goal is to learn a node representation $\mathbf{H}$ for downstairs tasks.

**First-order Graph ODE:** The first graph ODE method is proposed by (Xhonneux et al., 2020). Considering the Simple GNN (Wu et al., 2019) with $\mathbf{H}_{n+1} = \mathbf{A}\mathbf{H}_n + \mathbf{H}_0$, the solution is given by:

$$\frac{d\mathbf{H}(t)}{dt} = ln\mathbf{A}\mathbf{H}(t) + \mathbf{E}, \quad \mathbf{H}(t) = (\mathbf{A} - \mathbf{I})^{-1}(e^{(\mathbf{A}-\mathbf{I})t} - \mathbf{I})\mathbf{E} + e^{(\mathbf{A}-\mathbf{I})t}\mathbf{E}, \tag{1}$$

where $\mathbf{E} = \varepsilon(X)$ is the output of the encoder $\varepsilon$ and the initial value $\mathbf{H}(0) = (ln\mathbf{A})^{-1}(\mathbf{A} - \mathbf{I})\mathbf{E}$.

**Second-order Graph ODE:** To model high-order correlations in dynamic evolution, (Rusch et al., 2022) first propose the second-order graph ODE, which is represented as:

$$\mathbf{X}^{''} = \sigma(\mathbf{F}_\theta(\mathbf{X}, t)) - \gamma\mathbf{X} - \alpha\mathbf{X}^{'}, \tag{2}$$

where $(\mathbf{F}_\theta(\mathbf{X}, t))_i = \mathbf{F}_\theta(\mathbf{X}_i(t), \mathbf{X}_j(t), t)$ is a learnable coupling function with parameters $\theta$. Due to the unavailability of an analytical solution for Eq. 2, GraphCON (Rusch et al., 2022) addresses it through an iterative numerical solver employing a suitable time discretization method. GraphCON utilizes the IMEX (implicit-explicit) time-stepping scheme, an extension of the symplectic Euler method (Hairer et al., 1993) that accommodates systems with an additional damping term.

$$\mathbf{Y}^n = \mathbf{Y}^{n-1} + \Delta t \left[\sigma(\mathbf{F}_\theta(\mathbf{X}^{n-1}, t^{n-1})) - \gamma\mathbf{X}^{n-1} - \alpha\mathbf{Y}^{n-1}\right],$$
$$\mathbf{X}^n = \mathbf{X}^{n-1} + \Delta t\mathbf{Y}^n, \, n = 1, \cdots, N, \tag{3}$$

where $\Delta t > 0$ is a fixed time-step and $\mathbf{Y}^n, \mathbf{X}^n$ denote the hidden node features at time $t^n = n\Delta t$.

## 3.2 SPIKING NEURAL NETWORKS

**First-order SNNs:** In contrast to traditional artificial neural networks, SNNs convert input data into binary spikes over time, with each neuron in the SNNs maintaining a membrane potential that accumulates input spikes. A spike is produced as an output when the membrane potential exceeds a threshold. And the first-order SNNs is formulated as:

$$u_{\tau+1,i} = \lambda(u_{\tau,i} - V_{th}s_{\tau,i}) + \sum_j w_{ij}s_{\tau,j} + b, \quad s_{\tau+1,i} = \mathbb{H}(u_{\tau+1,i} - V_{th}), \tag{4}$$

where $\mathbb{H}(x)$ is the Heaviside function, which is the non-differentiable spiking function. $s_{\tau,i}$ is the binary spike train of neuron $i$, $\lambda$ is the constant. $w_{ij}$ and $b$ are the weights and bias of each neuron.

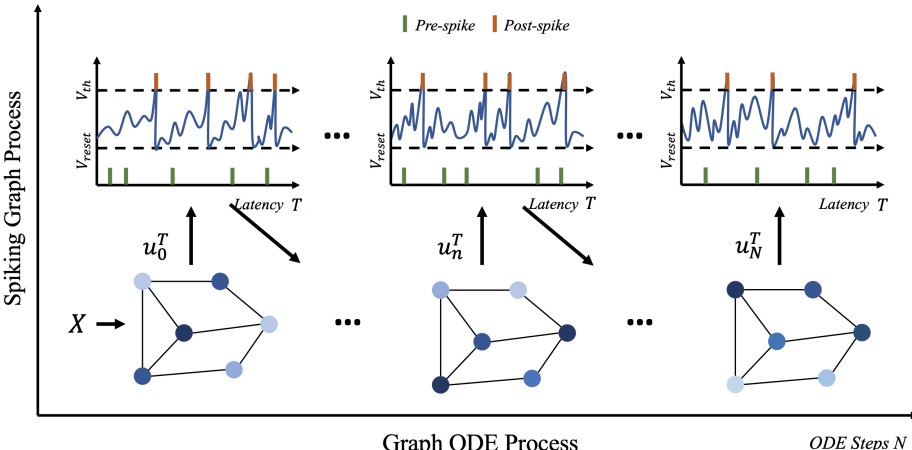

Figure 2: Overview of the proposed CSGO. The proposed CSGO takes a graph with node features as input, which are initially encoded using the SGNs (first-order or second-order). Subsequently, a high-order Graph ODE process is employed to evolve the dynamic representation of nodes. Finally, the representation is projected for downstream tasks.

**Second-order SNNs:** The first-order neuron models assume that an input voltage spike causes an immediate change in synaptic current, affecting the membrane potential. However, in reality, a spike leads to the gradual release of neurotransmitters from the pre-synaptic neuron to the post-synaptic neuron. To capture the temporal dynamics, we utilize the synaptic conductance-based LIF model, which considers the gradual changes in input current over time. To solve this, (Eshraghian et al., 2023) propose the second-order SNN, which is formulated as:

$$I_{\tau+1} = \alpha I_\tau + W X_{\tau+1}, \quad u_{\tau+1,i} = \beta u_{\tau,i} + I_{\tau+1,i} - R, \quad s_{\tau,i} = \mathbb{H}(u_{\tau+1,i} - V_{th}), \quad (5)$$

where $\alpha = exp(-\Delta t/\tau_{syn})$, $\beta = exp(-\Delta t/\tau_{mem})$, $\tau_{syn}$ models the time constant of the synaptic current in an analogous way to how $\tau_{mem}$ models the time constant of the membrane potential.

## 4 METHODOLOGY

In this part, we present the proposed CSGO for dynamic spiking graph learning. CSGO incorporates Graph ODE with SGNs into a unified framework, which preserves the advantage of Graph ODE for low energy consumption dynamic evolution. To mitigate the problem of information loss attributed to SGNs, we involve the derivation of second-order spike representation and differentiation for second-order SGNs, and then coordinate with high-order Graph ODE, referred to as CSGO-2nd. Finally, we present a theoretical proof to ensure that CSGO effectively mitigates the challenges associated with gradient exploding and vanishing. The details of CSGO are illustrated in Figure 2.

### 4.1 FIRST-ORDER CSGO

Specifically, SGNs propagate information within time latency $\tau$, and the Graph ODE evaluates feature evolution on different layers $l$. We propose the first-order CSGO, which integrates SGNs with Graph ODEs, allowing information to be interactively propagated through both SGNs and the Graph ODE:

**Proposition 1** *Define the first-order SNNs as $\frac{du_n^\tau}{d\tau} = g(u_n^\tau, \tau)$, and first-order Graph ODE as $\frac{du_n^\tau}{dn} = f(u_n^\tau, n)$, then the first-order CSGO can be formulated as:*

$$u_N^T = 2 \int_0^{N-1} f\left(\int_0^T g(u_y^x, x)dx\right) dy + \int_{N-1}^N f\left(\int_0^T g(u_y^x, x)dx\right) dy \quad (6)$$

$$= 2 \int_0^T g\left(\int_0^{N-1} f(u_y^x, x)dy\right) dx + \int_{N-1}^N f\left(\int_0^T g(u_y^x, x)dx\right) dy. \quad (7)$$

*where $T$ is the total latency of SNNs, and $N$ is the steps of Graph ODE, $u_y^x$ denotes the neuron membrane on latency $x \in [0, T]$ and ODE step $y \in [0, N]$. The derivation is shown in Appendix A.*

From Proposition 1, we observe that incorporating SNNs with Graph ODEs essentially involves evaluating the membrane potential $u$ through the ODE process and obtaining the spiking representation at each ODE step $n$. To model the dynamic process of graphs in spiking scenarios, CSGO-1st leverages the Graph ODE instead of calculating the spiking representation at every step, thereby efficiently addressing the issue of energy consumption. In our implementation of the CSGO-1st, we employ Eq. 15 by initially calculating spike representations with the initial real-valued features, followed by modeling the evolution of node embeddings. As described in (Meng et al., 2022), the first-order spike representation at step $n$ is denoted as: $\mathbf{H}(0) = \frac{\sum_{\tau=1}^{N} \lambda^{N-\tau} s_\tau^0}{\sum_{\tau=1}^{N} \lambda^{N-\tau}}$. By combining Eq. 1, we have:

$$\frac{d\mathbf{H}(n)}{dt} = ln\mathbf{A}\mathbf{H}(n) + \frac{\sum_{\tau=1}^{N} \lambda^{N-\tau} s_\tau^0}{\sum_{\tau=1}^{N} \lambda^{N-\tau}}, \tag{8}$$

where $s_\tau^0$ is the binary spiking representation on latency $\tau$ at the first step, and $\lambda = exp(-\frac{\Delta n}{\kappa})$ with $\Delta n \ll \kappa$, $\kappa$ is the time constant. We can then obtain the spiking output $s_n^T = \mathbb{H}(\mathbf{H}(n))$ on step $n$ with the Heaviside function $\mathbb{H}$, and utilize it for the final prediction.

## 4.2 SECOND-ORDER SPIKING NEURAL NETWORKS

The proposed first-order CSGO addresses the challenge of combining SNNs with Graph ODE to achieve energy-efficient modeling for dynamic graph learning. However, the first-order SNNs typically suffer from the information loss issue (Yin et al., 2023b). Motivated by recent advancements in high-order models (Luo et al., 2023), which addresses high-order correlations of nodes, we introduce the second-order CSGO to tackle the aforementioned issue. In this part, we begin by deriving the second-order spike representation and investigating the backpropagation of second-order SNNs.

### 4.2.1 SECOND-ORDER SNNs FORWARD PROPAGATION

We first propose the forward propagation of second-order SNNs. We set the forward propagation layer of SNNs to $L$. According to Eq. 5, the propagation can be formulated as:

$$u^i(\tau) = \beta^i u^i(\tau - 1) + (1 - \beta^i)\frac{V_{th}^{i-1}}{\Delta t}\left(\alpha^i I^{i-1}(\tau - 1) + \mathbf{W}^i s^{i-1}(\tau)\right) - V_{th}^i s^i(\tau),$$

where $i = 1, \cdots, L$ denotes the $i$-th layer, $s^0$ and $s^i$ denote the input and output of SNNs, respectively. $I^i$ is the input of the $i$-th layer, $\tau = 1, \cdots, T$ is the time step on SNNs and $T$ is the total latency. $\alpha^i = exp(-\Delta\tau/\tau_{syn}^i)$, $\beta^i = exp(-\Delta\tau/\tau_{mem}^i)$ and $0 < \Delta\tau \ll \{\tau_{syn}^i, \tau_{mem}^i\}$.

### 4.2.2 SECOND-ORDER SPIKE REPRESENTATION

Considering the second-order SNNs model defined by Eq. 5, we first define the weighted average input current as $\hat{I}(T) = \frac{1}{(\beta-\alpha)^2}\frac{\sum_{n=0}^{T-1}(\beta^{T-n}-\alpha^{T-n})I_{in}(n)}{\sum_{n=0}^{T-1}(\beta^{T-n}-\alpha^{T-n})}$, and the scaled weighted firing rate as $\hat{a}(T) = \frac{1}{\beta^2}\frac{V_{th}\sum_{n=0}^{T-1}\beta^{T-n}s(n)}{\sum_{n=0}^{T-1}(\beta^{T-n}-\alpha^{T-n})\Delta\tau}$. Here we treat $\hat{a}(T)$ as the spike train of $\{s(n)\}_{n=1}^T$. Similarly to the first-order spike representation (Meng et al., 2022), we directly determine the relationship between $\hat{I}(T)$ and $\hat{a}(T)$ using a differentiable mapping. Specifically, by combing Eq. 5, we have:

$$\begin{aligned} u(\tau + 1) =& \beta u(\tau) + \alpha I_{syn}(\tau) + I_{input}(\tau) - V_{th}s(\tau) \\ =& \beta^2 u(\tau - k + 1) + \alpha\sum_{i=0}^{k-1}\beta^i I_{syn}(\tau - i) + \sum_{i=0}^{k-1}\beta^i(I_{input}(\tau - i) - V_{th}s(\tau - i)). \end{aligned} \tag{9}$$

By summing Eq. 9 over $\tau = 1$ to $T$, we have:

$$u(T) = \frac{1}{\beta - \alpha}\sum_{n=0}^{T-1}(\beta^{T-n} - \alpha^{T-n})I_{in}(n) - \frac{1}{\beta}\sum_{n=0}^{T-1}\beta^{T-n}V_{th}s(n). \tag{10}$$

Dividing Eq. 10 by $\Delta\tau \sum_{n=0}^{T-1}(\beta^{T-n} - \alpha^{T-n})$:

$$\hat{a}(T) = \frac{\beta - \alpha}{\beta}\frac{\hat{I}(T)}{\Delta\tau} - \frac{u(T)}{\Delta\tau\beta\sum_{n=0}^{T-1}(\beta^{T-n} - \alpha^{T-n})}$$

$$\approx \frac{\tau_{syn}\tau_{mem}}{\tau_{mem} - \tau_{syn}}\hat{I}(T) - \frac{u(T)}{\Delta\tau\beta\sum_{n=0}^{T-1}(\beta^{T-n} - \alpha^{T-n})},$$

since $\lim_{\Delta\tau\to 0}\frac{1-\alpha/\beta}{\Delta\tau} = \frac{\tau_{syn}\tau_{mem}}{\tau_{mem}-\tau_{syn}}$ and $\Delta\tau \ll \frac{1}{\tau_{syn}} - \frac{1}{\tau_{mem}}$, we can approximate $\frac{\beta-\alpha}{\beta\Delta\tau}$ by $\frac{\tau_{syn}\tau_{mem}}{\tau_{mem}-\tau_{syn}}$. Following (Meng et al., 2022), and take $\hat{a}(T) \in [0, \frac{V_{th}}{\Delta\tau}]$ into consideration and assume $V_{th}$ is small, we ignore the term $\frac{u(T)}{\Delta\tau\beta\sum_{n=0}^{T-1}(\beta^{T-n}-\alpha^{T-n})}$, and approximate $\hat{a}(T)$ with:

$$\lim_{T\to\infty}\hat{a}(T) \approx clamp\left(\frac{\tau_{syn}\tau_{mem}}{\tau_{mem} - \tau_{syn}}\hat{I}(T), 0, \frac{V_{th}}{\Delta\tau}\right), \tag{11}$$

where $clamp(x, a, b) = max(a, min(x, b))$. During the training of the second-order SNNs, we have Proposition 2, and the detailed derivation is shown in Appendix B.

**Proposition 2** *Define* $\hat{a}^0(T) = \frac{\sum_{n=0}^{T-1}\beta_i^{T-n-2}s^0(n)}{\sum_{n=0}^{T-1}(\beta_i^{T-n}-\alpha_i^{T-n})\Delta\tau}$ *and* $\hat{a}^i(T) = \frac{V_{th}^i\sum_{n=0}^{T-1}\beta_i^{T-n-2}s^i(n)}{\sum_{n=0}^{T-1}(\beta_i^{T-n}-\alpha_i^{T-n})\Delta\tau}$, $i \in [1, L]$, *where* $\alpha^i = exp(-\Delta\tau/\tau_{syn}^i)$ *and* $\beta^i = exp(-\Delta\tau/\tau_{mem}^i)$. *The differentiable mappings is:*

$$\mathbf{z}^i = clamp\left(\frac{\tau_{syn}^i\tau_{mem}^i}{\tau_{mem}^i - \tau_{syn}^i}\mathbf{W}^i\mathbf{z}^{i-1}, 0, \frac{V_{th}^i}{\Delta\tau}\right), i = 1, \cdots, L.$$

*If* $\lim_{T\to\infty}\hat{a}^i(T) = \mathbf{z}^i$ *for* $i = 0, 1, \cdots, L-1$, *then* $\hat{a}^{i+1}(T) \approx \mathbf{z}^{i+1}$ *when* $T \to \infty$

### 4.2.3 DIFFERENTIATION ON SECOND-ORDER SPIKE REPRESENTATION

In this part, we use the spike representation to drive the backpropagation training algorithm for second-order SNNs. With the forward propagation of the $i$-th layers, we get the output of SNN with $s^i = \{s^i(1), \cdots, s^i(T)\}$, $i \in [1, L]$. We define the spike representation operator $r(s) = \frac{1}{\beta^2}\frac{V_{th}\sum_{n=0}^{T-1}\beta^{T-n}s(n)}{\sum_{n=0}^{T-1}(\beta^{T-n}-\alpha^{T-n})\Delta\tau}$, and get the final output $\mathbf{o}^L = r(s^L)$. For the simple second-order SNN, assuming the loss function as $\mathcal{L}$, we calculate the gradient $\frac{\partial\mathcal{L}}{\partial\mathbf{W}^i}$ as:

$$\frac{\partial\mathcal{L}}{\partial\mathbf{W}^i} = \frac{\partial\mathcal{L}}{\partial\mathbf{o}^i}\frac{\partial\mathbf{o}^i}{\partial\mathbf{W}^i} = \frac{\partial\mathcal{L}}{\partial\mathbf{o}^{i+1}}\frac{\partial\mathbf{o}^{i+1}}{\partial\mathbf{o}^i}\frac{\partial\mathbf{o}^i}{\partial\mathbf{W}^i}, \quad \mathbf{o}^i = r(s^i) \approx clamp\left(\mathbf{W}^ir(s^{i-1}), 0, \frac{V_{th}^i}{\Delta\tau}\right). \tag{12}$$

We can compute the gradient of second-order SNNs by calculating $\frac{\partial\mathbf{o}^{i+1}}{\partial\mathbf{o}^i}$ and $\frac{\partial\mathbf{o}^i}{\partial\mathbf{W}^i}$ based on Eq. 12.

## 4.3 SECOND-ORDER CSGO

Having obtained the second-order spike representation for SNNs, we introduce the second-order CSGO. While obtaining an analytical solution for the second-order CSGO may not be feasible, we can derive a conclusion similar to Proposition 1. The specifics are presented as follows.

**Proposition 3** *Define the second-order SNNs as* $\frac{d^2u_t^\tau}{d\tau^2} + \delta\frac{du_t^\tau}{d\tau} = g(u_t^\tau, \tau)$, *and second-order Graph ODE as* $\frac{d^2u_t^\tau}{dt^2} + \gamma\frac{du_t^\tau}{dt} = f(u_t^\tau, t)$, *then the second-order CSGO follows:*

$$u_t^\tau = \int_0^N h\left(\int_0^T e(u_t^\tau)d\tau\right)dt = \int_0^T e\left(\int_0^N h(u_t^\tau)dt\right)d\tau,$$

$$s.t. \quad \frac{\partial^2u_t^\tau}{\partial\tau^2} + \delta\frac{\partial u_t^\tau}{\partial\tau} = g(u_t^\tau), \quad \frac{\partial^2u_t^\tau}{\partial t^2} + \gamma\frac{\partial u_t^\tau}{\partial t} = f(u_t^\tau),$$

*where* $e(u_t^\tau) = \int_0^T g(u_t^\tau)d\tau - \delta(u_t^T - u_t^0)$, $h(u_t^\tau) = \int_0^N f(u_t^\tau)dt - \gamma(u_N^\tau - u_0^\tau)$, $\frac{\partial e(u_t^\tau)}{\partial\tau} = g(u_t^\tau)$ *and* $\frac{\partial h(u_t^\tau)}{\partial t} = f(u_t^\tau)$. $\delta$ *and* $\gamma$ *are the hyperparameters of second-order SNNs and Graph ODE.*

The details are derived in Appendix C. Similarly to the CSGO-1st, we implement the CSGO-2nd by calculating the spike representation on the initial step with Eq. 11 and then modeling the evolution of node embeddings with second-order Graph ODE using Eq. 3. Furthermore, we analyze the differentiation of the CSGO-2nd to optimize its performance. Denote the loss function as $\mathcal{L} = \sum_{i \in \mathcal{V}} \left| \mathbf{X}_i^T - \bar{\mathbf{X}}_i \right|^2$, and $\bar{\mathbf{X}}_i$ is the label of node $i$. With the chain rule, we have: $\frac{\partial \mathcal{L}}{\partial \mathbf{W}^l} = \frac{\partial \mathcal{L}}{\partial \mathbf{o}_N^T} \frac{\partial \mathbf{o}_N^T}{\partial \mathbf{o}_N^l} \frac{\partial \mathbf{o}_T^l}{\partial \mathbf{W}^l}$.
As traditional GNN models face the problem of exploding or vanishing gradients (Rusch et al., 2022), we further analyze the upper bound of the gradient in the proposed CSGO-2nd.

**Proposition 4** *Let $\mathbf{X}^n$ and $\mathbf{Y}^n$ be the node features, generated by Eq. 3, and $\Delta t \ll 1$. The gradients of the second-order Graph ODE $\mathbf{W}_l$ and second-order SNNs $\mathbf{W}^k$ are bounded as follows:*

$$\left| \frac{\partial \mathcal{L}}{\partial \mathbf{W}_l} \right| \leq \frac{\beta' \hat{\mathbf{D}} \Delta t (1 + \Gamma N \Delta t)}{v} \left( \max_{1 \leq i \leq v} (|\mathbf{X}_i^0| + |\mathbf{Y}_i^0|) \right)$$
$$+ \frac{\beta' \hat{\mathbf{D}} \Delta t (1 + \Gamma N \Delta t)}{v} \left( \max_{1 \leq i \leq v} |\bar{\mathbf{X}}_i| + \beta \sqrt{N \Delta t} \right)^2, \tag{13}$$

$$\left| \frac{\partial \mathcal{L}}{\partial \mathbf{W}^k} \right| \leq \frac{(1 + N \Gamma \Delta t)(1 + L \Theta \Delta \tau) V_{th}}{v \beta^2 \Delta \tau} \left( \max_{1 \leq i \leq v} |\mathbf{X}_i^N| + \max_{1 \leq i \leq v} |\bar{\mathbf{X}}_i| \right). \tag{14}$$

*where $\beta = \max_x |\sigma(x)|$, $\beta' = \max_x |\sigma'(x)|$, $\hat{D} = \max_{i,j \in \mathcal{V}} \frac{1}{\sqrt{d_i d_j}}$, and $\Gamma := 6 + 4\beta' \hat{D} \max_{1 \leq n \leq T} ||\mathbf{W}^n||_1$, $\Theta := 6 + 4\beta' \hat{D} \max_{1 \leq n \leq N} ||\mathbf{W}^n||_1$. $d_i$ is the degree of node $i$, $\bar{\mathbf{X}}_i$ is the label of node $i$. Eq. 13 can be obtained from (Rusch et al., 2022) directly, and the derivation of the Eq. 14 is presented in Appendix D.*

The upper bound in Proposition 4 demonstrates that the total gradient remains globally bounded, regardless of the number of Graph ODE layers $N$ and SNNs layers $L$, as long as $\Delta t \sim N^{-1}$ and $\Delta \tau \sim L^{-1}$. This effectively addresses the issues of exploding and vanishing gradients.

## 5 EXPERIMENTS

To evaluate the effectiveness of our proposed CSGO, we conduct extensive experiments with CSGO across various graph learning tasks, including node classification and graph classification.

### 5.1 EXPERIMENTAL SETTINGS

**Datasets.** For the node classification, we evaluate CSGO on homophilic (i.e., Cora (McCallum et al., 2000), Citeseer (Sen et al., 2008) and Pubmed (Namata et al., 2012)) and heterophilic (i.e., Texas, Wisconsin and Cornell from the WebKB[1]) datasets, where high homophily indicates that a node's features are similar to those of its neighbors, and heterophily suggests the opposite. The homophily level is measured according to (Pei et al., 2020), and is reported in Table 1 and 2. In the graph classification task, we utilize the MNIST dataset (LeCun et al., 1998). To represent the grey-scale images as irregular graphs, we associate each superpixel (large blob of similar color) with a vertex, and the spatial adjacency between superpixels with edges. Each graph consists of a fixed number of 75 superpixels (vertices). To ensure consistent evaluation, we adopt the standard splitting of 55K-5K-10K for training, validation, and testing purposes (Rusch et al., 2022).

**Baselines.** For the homophilic datasets, we use standard GNN baselines: GCN (Kipf & Welling, 2017), SGC (Wu et al., 2019), GAT (Velickovic et al., 2017), MoNet (Monti et al., 2017), Graph-Sage (Hamilton et al., 2017), CGNN (Xhonneux et al., 2020), GDE (Poli et al., 2019), GRAND (Chamberlain et al., 2021), GraphCON (Rusch et al., 2022) and SpikingGCN (Zhu et al., 2022). Due to the assumption that neighbor feature similarity does not hold in heterophilic datasets, we utilize additional GNNs as baselines: GPRGNN (Chien et al., 2020), H2GCN (Zhu et al., 2020a), GCNII (Chen et al., 2020), Geom-GCN (Pei et al., 2020) and PairNorm (Zhao & Akoglu, 2019). For graph classification task, we apply ChebNet (Defferrard et al., 2016), PNCNN (Finzi et al., 2021), SplineCNN (Fey et al., 2018), GIN (Xu et al., 2019), and GatedGCN (Bresson & Laurent, 2017) for comparison.

---

[1]http://www.cs.cmu.edu/afs/cs.cmu.edu/project/theo-11/www/wwkb/

Table 1: The test accuracy (in %) for node classification on homophilic datasets. The results are calculated by averaging the results of 20 random initializations across 5 random splits. The mean and standard deviation of these results are obtained. **Bold** numbers means the best performance, and underline numbers indicates the second best performance.

| Homophily level | Cora 0.81 | Citeseer 0.74 | Pubmed 0.80 |
|---|---|---|---|
| GAT-ppr | 81.6±0.3 | 68.5±0.2 | 76.7±0.3 |
| MoNet | 81.3±1.3 | 71.2±2.0 | 78.6±2.3 |
| GraphSage | 79.2±7.7 | 71.6±1.9 | 77.4±2.2 |
| CGNN | 81.4±1.6 | 66.9±1.8 | 66.6±4.4 |
| GDE | 78.7±2.2 | 71.8±1.1 | 73.9±3.7 |
| GCN | 81.5±1.3 | 71.9±1.9 | 77.8±2.9 |
| GAT | 81.8±1.3 | 71.4±1.9 | 78.7±2.3 |
| SGC | 81.5±0.4 | 71.7±0.4 | 79.2±0.3 |
| GRAND | 83.6±1.0 | 73.4±0.5 | 78.8±1.7 |
| GraphCON-GCN | 81.9±1.7 | 72.9±2.1 | 78.8±2.6 |
| GraphCON-GAT | 83.2±1.4 | 73.2±1.8 | 79.5±1.8 |
| SpikingGCN | 80.7±0.6 | 72.5±0.2 | 77.6±0.5 |
| CSGO-1st | 83.3±2.1 | 73.7±2.0 | 76.9±2.7 |
| CSGO-2nd | **83.7±1.3** | **75.2±2.0** | **79.6±2.3** |

Table 2: The test accuracy (in %) for node classification on heterophilic datasets. All results represent the average performance of the respective model over 10 fixed train/val/test splits. **Bold** numbers means the best performance, and underline numbers indicates the second best performance.

| Homophily level | Texas 0.11 | Wisconsin 0.21 | Cornell 0.30 |
|---|---|---|---|
| GPRGNN | 78.4±4.4 | 82.9±4.2 | 80.3±8.1 |
| H2GCN | 84.9±7.2 | 87.7±5.0 | 82.7±5.3 |
| GCNII | 77.6±3.8 | 80.4±3.4 | 77.9±3.8 |
| Geom-GCN | 66.8±2.7 | 64.5±3.7 | 60.5±3.7 |
| PairNorm | 60.3±4.3 | 48.4±6.1 | 58.9±3.2 |
| GraphSAGE | 82.4±6.1 | 81.2±5.6 | 76.0±5.0 |
| MLP | 80.8±4.8 | 85.3±3.3 | 81.9±6.4 |
| GCN | 55.1±5.2 | 51.8±3.1 | 60.5±5.3 |
| GAT | 52.2±6.6 | 49.4±4.1 | 61.9±5.1 |
| GraphCON-GCN | 85.4±4.2 | 87.8±3.3 | **84.3±4.8** |
| GraphCON-GAT | 82.2±4.7 | 85.7±3.6 | 83.2±7.0 |
| CSGO-1st | 81.6±6.2 | 84.9±3.2 | 80.4±1.9 |
| CSGO-2nd | **87.3±4.2** | **88.8±2.5** | 83.7±2.7 |

**Implementation Details.** For the homophilic node classification task, we report the average results of 20 random initialization across 5 random splits. For the heterophilic node classification task, we present the average performance of the respective model over 10 fixed train/val/test splits. The results of baselines are reported in (Rusch et al., 2022). For CSGO-1st, we set the hyperparameter $\lambda$ to 1. As for CSGO-2nd, we set the hyperparameters $\alpha$ and $\beta$ to 1 as default. The time latency $N$ in SNNs are set to 8. For all the methods, we set the hidden size to 64 and the learning rate to 0.001 as default. All the experiments are conducted on the same device, equipped with NVIDIA A6000 GPU.

## 5.2 PERFORMANCE COMPARISION

**Homophilic Node Classification**. Table 1 shows the results of the proposed CSGO with the comparison of baselines. From the results, we find that: (1) Compared with the discrete methods (i.e., the baselines excluding Graph-CON), the continuous methods (GraphCON and CSGO) achieve the best and second best performance, indicating that the continuous methods would help to capture the dynamic changes and subtle dynamics from graphs. (2) CSGO-1st and CSGO-2nd outperforms other baselines in most cases. We attribute that, even if SNNs loses some detailed information, CSGO can still achieve good performance on the relatively simple homophilic dataset. Furthermore, the application of SNNs contributes to improved efficiency in the CSGO framework. (3) CSGO-2nd consistently outperforms the CSGO-1st. This

Table 3: The test accuracy (in %) for graph classification on MNIST datasets. **Bold** numbers means the best performance, and underline numbers indicates the second best performance.

| Model | Test accuracy |
|---|---|
| ChebNet (Defferrard et al., 2016) | 75.62 |
| MoNet (Monti et al., 2017) | 91.11 |
| PNCNN (Finzi et al., 2021) | 98.76 |
| SplineCNN (Fey et al., 2018) | 95.22 |
| GIN (Xu et al., 2019) | 97.23 |
| GatedGCN (Bresson & Laurent, 2017) | 97.95 |
| GCN (Kipf & Welling, 2017) | 88.89 |
| GAT (Velickovic et al., 2017) | 96.19 |
| GraphCON-GCN (Rusch et al., 2022) | 98.68 |
| GraphCON-GAT (Rusch et al., 2022) | 98.91 |
| CSGO-1st | 98.82 |
| CSGO-2nd | **98.92** |

highlights the significance of introducing high-order structures to preserve information and mitigates the information loss issue caused by first-order SNNs. Although high-order structures suffer higher energy costs compared to first-order, the performance gains make it worthwhile to deploy them. (4) CSGO-1st and CSGO-2nd outperforms the spiking-based method (i.e., Spiking) in most case. This can be attributed to the incorporation of Graph ODE, which efficiently captures the dynamic evolution while maintaining low energy consumption.

**Heterophilic Node Classification**. Table 2 shows the results of heterophilic node classification, and we observe that: (1) The traditional message-passing-based methods (GCN, GAT, GraphSAGE and Geom-GCN) perform worse than the well-designed methods (GPRGNN, H2GCN, GCNII, GraphCON and CSGO) for heterophilic datasets. This disparity comes from the inaccurate assumption of neighbor feature similarity, which doesn't hold in heterophilic datasets. The propagation of heterophilic information between nodes would degrade the model's representation ability, leading to a decline in performance. (2) The CSGO-1st performs less effectively than GraphCON. This is because node prediction tasks on heterophilic datasets are more influenced by the characteristics of heterophilic features compared to homophilic datasets. Consequently, the information loss issue caused by first-order SNNs results in worse model performance. (3) The CSGO-2nd consistently outperforms CSGO-1st, providing further evidence of the effectiveness of high-order structures in preserving information and mitigating the issue of information loss.

**Graph Classification**. We present the graph classification results of our proposed CSGO alongside comparison baselines in Table 3. From the results, we have the following observations: (1) In the graph classification tasks, dynamic graph methods (i.e., CSGO and GraphCON) consistently outperform the baseline methods across all cases. This underscores the importance of employing a continuous processing approach when dealing with graph data, enabling the extraction of continuous changes and subtle dynamics from graphs. (2) The CSGO-1st performs worse than the CSGO-2nd, highlighting the significance of incorporating high-order structures to obtain additional information for prediction, without incurring significant overhead. (3) The CSGO-1st performs worse than GraphCON-GAT and better than GraphCON-GCN. Compared to GraphCON-GCN, the information loss caused by SNNs does not critically affect graph representation ability. On the contrary, the binarization operation of SNNs contributes to reduced energy consumption. Graph-GAT outperforms CSGO-1st, mainly because the GAT method enhances graph representation. However, Graph-GAT still lags behind CSGO-2nd, indicating that the introduction of high-order structures mitigates the information loss issue associated with first-order methods.

## 5.3 Energy Efficiency Analysis

To assess the energy efficiency of CSGO, we use the metric from (Zhu et al., 2022), which quantifies the energy consumption for node prediction. Specifically, we follow the spike method (Cao et al., 2015), counting the total spikes during inference across three datasets to estimate the energy consumption of SNNs. In Figure 3, we compare the energy consumption of traditional GNNs, including standard methods like GCN and GAT, Graph ODE methods such as GraphCon-GCN and GraphCon-GAT, the spike-based method SpikeGCN, and the proposed CSGO-1st and CSGO-2nd. Traditional GNNs are evaluated on GPUs (NVIDIA A6000), while, follow-

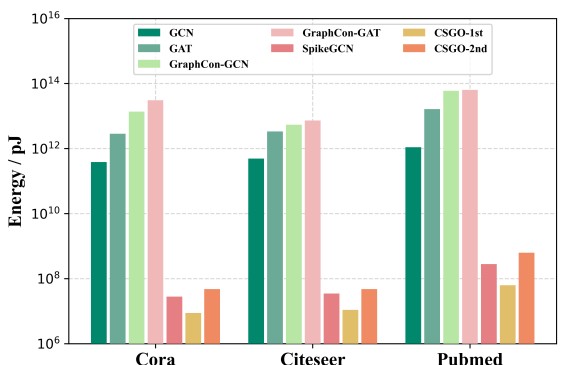

Figure 3: Energy consumption comparison with various baselines on different datasets.

ing (Zhu et al., 2022), the spike-based models are evaluated on neuromorphic chips (ROLLS (Indiveri et al., 2015)). From the results, we find that (1) The spike-based methods, i.e., SpikeGCN, CSGO-1st and CSGO-2nd, exhibit significantly lower energy consumption compared to traditional GNNs, demonstrating the superior energy efficiency of SNNs. (2) The CSGO-1st has a lower energy consumption than SpikeGCN, while CSGO-2nd consumes slightly more than SpikeGCN. Given the better performance of CSGO-2nd, it is worthwhile to deploy CSGO-2nd. Besides, we analyze the ablation study and hyperparameters of CSGO. The details are presented in Appendix E and F.

## 6 Conclusion

In this paper, we address the practical problem of continuous spiking graph learning and propose an effective method named CSGO. CSGO integrates SNNs and Graph ODE into a unified framework

from two distinct dimensions, thus retaining the benefits of low-power consumption and fine-grained feature extraction. Considering that the high-order structure would help to relieve the problem of information loss, we derive the second-order spike representation and investigate the backpropagation of second-order SNNs, by incorporating with high-order Graph ODE, we introduce the second-order CSGO. Furthermore, to ensure the stability of CSGO, we prove that CSGO mitigates the gradient exploding and vanishing problem. Extensive experiments on diverse datasets validate the efficacy of proposed CSGO compared with various competing methods. In future work, we will explore the higher-order structure for more efficient continuous graph learning.

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

## A    PROOF OF PROPOSITION 1

**Proposition 1** *Define the first-order SNNs as $\frac{du_n^\tau}{d\tau} = g(u_n^\tau, \tau)$, and first-order Graph ODE as $\frac{du_n^\tau}{dn} = f(u_n^\tau, n)$, then the first-order CSGO can be formulated as:*

$$u_N^T = 2 \int_0^{N-1} f\left( \int_0^T g(u_y^x, x)dx \right) dy + \int_{N-1}^N f\left( \int_0^T g(u_y^x, x)dx \right) dy \tag{15}$$

$$= 2 \int_0^T g\left( \int_0^{N-1} f(u_y^x, x)dy \right) dx + \int_{N-1}^N f\left( \int_0^T g(u_y^x, x)dx \right) dy. \tag{16}$$

*where $T$ is the total latency of SNNs, and $N$ is the steps of Graph ODE, $u_y^x$ denotes the neuron membrane on latency $x \in [0, T]$ and ODE step $y \in [0, N]$.*

*Proof.*

$$\frac{du_n^\tau}{d\tau} = g(u_n^\tau, \tau), \quad \frac{du_n^\tau}{dn} = f(u_n^\tau, n),$$

$u_n^\tau$ is a function related to variable $n$ and $\tau$, thus,

$$u_n^{\tau+1} = u_n^\tau + \int_\tau^{\tau+1} g(u_n^x, x)dx, \;\; u_{n+1}^{\tau+1} = u_n^{\tau+1} + \int_n^{n+1} f(u_y^{\tau+1}, y)dy, \tag{17}$$

$$u_N^T = u_{N-1}^{T-1} + \int_{N-1}^N f(u_y^T, y)dy + \int_{T-1}^T g(u_{N-1}^x, x)dx$$

$$= u_{N-2}^{T-2} + \int_{N-2}^N f\left( u_y^T, y \right) dy + \int_{T-2}^T g(u_{N-1}^x, x)dx$$

$$= u_0^0 + \int_0^N f\left( u_y^T, y \right) dy + \int_0^T g(u_{N-1}^x, x)dx$$

$$= u_0^0 + \int_0^N f\left( u_y^{T-1} + \int_{T-1}^T g(u_y^x, x)dx \right) dy + \int_0^T g\left( u_{N-2}^x + \int_{N-2}^{N-1} f(u_y^x, y)dy \right) dx$$

$$= u_0^0 + \int_0^N f\left( u_y^0 + \int_0^T g(u_y^x, x)dx \right) dy + \int_0^T g\left( u_0^x + \int_0^{N-1} f(u_y^x, y)dy \right) dx. \tag{18}$$

By adding the initial state on each time step and latency with $u_t^0 = 0$ and $u_0^\tau = 0$, we have:

$$u_N^T = \int_0^N f\left( \int_0^T g(u_y^x, x)dx \right) dy + \int_0^T g\left( \int_0^{N-1} f(u_y^x, y)dy \right) dx$$

$$= \underbrace{\int_0^{N-1} f\left( \int_0^T g(u_y^x, x)dx \right) dy + \int_0^T g\left( \int_0^{N-1} f(u_y^x, y)dy \right) dx}_{first\ term} + \underbrace{\int_{N-1}^N f\left( \int_0^T g(u_y^x, x)dx \right) dy}_{second\ term}$$

$$= 2 \int_0^{N-1} f\left( \int_0^T g(u_y^x, x)dx \right) dy + \int_{N-1}^N f\left( \int_0^T g(u_y^x, x)dx \right) dy. \tag{19}$$

The first term denotes that the SNNs and Graph ODE are interactively updated during the time step $0$ to $T-1$, and the second term denotes that at the last step $T$, CSGO simply calculates the Graph ODE process while ignoring the SNNs for prediction.

## B  PROOF OF PROPOSITION 2

**Proposition 2** *Define* $\hat{a}^0(T) = \frac{\sum_{n=0}^{T-1} \beta_i^{T-n-2} s^0(n)}{\sum_{n=0}^{T-1} (\beta_i^{T-n} - \alpha_i^{T-n}) \Delta\tau}$ *and* $\hat{a}^i(T) = \frac{V_{th}^i \sum_{n=0}^{T-1} \beta_i^{T-n-2} s^i(n)}{\sum_{n=0}^{T-1} (\beta_i^{T-n} - \alpha_i^{T-n}) \Delta\tau}$, $i \in [1, L]$, *where* $\alpha^i = exp(-\Delta\tau/\tau_{syn}^i)$ *and* $\beta^i = exp(-\Delta\tau/\tau_{mem}^i)$. *The differentiable mappings is:*

$$\mathbf{z}^i = clamp\left( \frac{\tau_{syn}^i \tau_{mem}^i}{\tau_{mem}^i - \tau_{syn}^i} \mathbf{W}^i \mathbf{z}^{i-1}, 0, \frac{V_{th}^i}{\Delta\tau} \right), i = 1, \cdots, L.$$

*If* $\lim_{T\to\infty} \hat{a}^i(T) = \mathbf{z}^i$ *for* $i = 0, 1, \cdots, L-1$, *then* $\hat{a}^{i+1}(T) \approx \mathbf{z}^{i+1}$ *when* $T \to \infty$.

*Proof.* From Eq. 9, we have:

$$u(\tau+1) = \beta^2 u(\tau - k + 1) + \alpha \sum_{i=0}^{k-1} \beta^i I_{syn}(\tau - i) + \sum_{i=0}^{k-1} \beta^i (I_{input}(\tau - i) - V_{th} s(\tau - i)), \quad (20)$$

$$u(T) = \alpha \sum_{n=0}^{T-1} \beta^n I_{syn}(T - n - 1) + \sum_{n=0}^{T-1} \beta^n (I_{input}(T - n - 1) - V_{th} s(T - n - 1)). \quad (21)$$

Due to:

$$I_{syn}(\tau + 1) = \alpha^k I_{syn}(\tau - k + 1) + \sum_{i=0}^{k} \alpha^i I_{input}(\tau - i), \quad (22)$$

we have,

$$u(T) = \alpha \sum_{n=0}^{T-1} \beta^{T-n-1} I_{syn}(n) + \sum_{n=0}^{T-1} \beta^{T-n-1} (I_{input}(n) - V_{th} s(n))$$

$$= \alpha \left( \left( \frac{\beta^{T-1} \alpha^{-1} \left(1 - (\frac{\alpha}{\beta})^T\right)}{1 - \frac{\alpha}{\beta}} \right) I_{in}(0) + \left( \frac{\beta^{T-2} \alpha^{-1} \left(1 - (\frac{\alpha}{\beta})^{T-1}\right)}{1 - \frac{\alpha}{\beta}} \right) I_{in}(1) + \cdots \right.$$

$$\left. + \left( \frac{\beta^{T-i} \alpha^{-1} \left(1 - (\frac{\alpha}{\beta})^{T-i+1}\right)}{1 - \frac{\alpha}{\beta}} \right) I_{in}(i - 1) + \cdots + (\beta^2 \alpha^{-1} + \beta + \alpha) I_{in}(T - 3) \right.$$

$$\left. + (\beta \alpha^{-1} + 1) I_{in}(T - 2) + \alpha^{-1} I_{in}(T - 1) \right) - \sum_{n=0}^{T-1} \beta^{T-n-1} V_{th} s(n)$$

$$= \frac{1}{\beta - \alpha} \left( \left( \beta^T \left(1 - \left(\frac{\alpha}{\beta}\right)^T\right) I_{in}(0) \right) + \cdots + \left( \beta^{T-i+1} \left(1 - \left(\frac{\alpha}{\beta}\right)^{T-i+1}\right) I_{in}(i - 1) \right) \right.$$

$$\left. + \cdots + (\beta - \alpha) I_{in}(T - 1) \right) - \sum_{n=0}^{T-1} \beta^{T-n-1} V_{th} s(n)$$

$$= \frac{1}{\beta - \alpha} \sum_{n=0}^{T-1} (\beta^{T-n} - \alpha^{T-n}) I_{in}(n) - \sum_{n=0}^{T-1} \beta^{T-n-1} V_{th} s(n).$$

Define $\hat{I}(T) = \frac{1}{(\beta-\alpha)^2} \frac{\sum_{n=0}^{T-1} (\beta^{T-n} - \alpha^{T-n}) I_{in}(n)}{\sum_{n=0}^{T-1} (\beta^{T-n} - \alpha^{T-n})}$, and $\hat{a}(T) = \frac{1}{\beta^2} \frac{V_{th} \sum_{n=0}^{T-1} \beta^{T-n} s(n)}{\sum_{n=0}^{T-1} (\beta^{T-n} - \alpha^{T-n})}$, we have:

$$\hat{a}(T) = \frac{\beta - \alpha}{\beta} \frac{\hat{I}(T)}{\Delta\tau} - \frac{u(T)}{\Delta\tau \beta \sum_{n=0}^{T-1} (\beta^{T-n} - \alpha^{T-n})} \approx \frac{\tau_{syn} \tau_{mem}}{\tau_{mem} - \tau_{syn}} \hat{I}(T) - \frac{u(T)}{\Delta\tau \beta \sum_{n=0}^{T-1} (\beta^{T-n} - \alpha^{T-n})},$$

where $\alpha = exp(-\Delta\tau/\tau_{syn})$, $\beta = exp(-\Delta\tau/\tau_{mem})$.

Following Meng et al. (2022), and take $\hat{a}(T) \in [0, \frac{V_{th}}{\Delta\tau}]$ into consideration and assume $V_{th}$ is small, we ignore the term $\frac{u(T)}{\Delta\tau \beta \sum_{n=0}^{T-1} (\beta^{T-n} - \alpha^{T-n})}$, and approximate $\hat{a}(T)$ with $clamp\left( \frac{\tau_{syn} \tau_{mem}}{\tau_{mem} - \tau_{syn}} \hat{I}(T), 0, \frac{V_{th}}{\Delta\tau} \right)$. Take the average input $\hat{I}(T) = \mathbf{W}\mathbf{z}$, we have $\mathbf{z}^i = clamp\left( \frac{\tau_{syn}^i \tau_{mem}^i}{\tau_{mem}^i - \tau_{syn}^i} \mathbf{W}^i \mathbf{z}^{i-1}, 0, \frac{V_{th}^i}{\Delta\tau} \right)$. If $\lim_{T\to\infty} \hat{a}^i(T) = \mathbf{z}^i$, then $\hat{a}^{i+1}(T) \approx \mathbf{z}^{i+1}$ when $T \to \infty$.

## C  PROOF OF PROPOSITION 3

**Proposition 3** *Define the second-order SNNs as $\frac{d^2 u_t^\tau}{d\tau^2} + \delta \frac{du_t^\tau}{d\tau} = g(u_t^\tau, \tau)$, and second-order Graph ODE as $\frac{d^2 u_t^\tau}{dt^2} + \gamma \frac{du_t^\tau}{dt} = f(u_t^\tau, t)$, then the second-order CSGO follows:*

$$u_t^\tau = \int_0^N h\left(\int_0^T e(u_t^\tau)d\tau\right)dt = \int_0^T e\left(\int_0^N h(u_t^\tau)dt\right)d\tau,$$

$$s.t. \quad \frac{\partial^2 u_t^\tau}{\partial \tau^2} + \delta\frac{\partial u_t^\tau}{\partial \tau} = g(u_t^\tau), \quad \frac{\partial^2 u_t^\tau}{\partial t^2} + \gamma\frac{\partial u_t^\tau}{\partial t} = f(u_t^\tau),$$

*where $e(u_t^\tau) = \int_0^T g(u_t^\tau)d\tau - \delta(u_t^T - u_t^0)$, $h(u_t^\tau) = \int_0^N f(u_t^\tau)dt - \gamma(u_N^\tau - u_0^\tau)$, $\frac{\partial e(u_t^\tau)}{\partial \tau} = g(u_t^\tau)$ and $\frac{\partial h(u_t^\tau)}{\partial t} = f(u_t^\tau)$. $\delta$ and $\gamma$ are the hyperparameters of second-order SNNs and Graph ODE.*

*Proof.* Obviously,

$$\frac{\partial^2 u_t^\tau}{\partial \tau^2} + \delta\frac{\partial u_t^\tau}{\partial \tau} = g(u_t^\tau), \quad \frac{\partial^2 u_t^\tau}{\partial t^2} + \gamma\frac{\partial u_t^\tau}{\partial t} = f(u_t^\tau),$$

$$so, \quad \frac{\partial u_t^\tau}{\partial \tau} + \delta(u_t^T - u_t^0) = \int_0^T g(u_t^\tau)d\tau, \quad \frac{\partial u_t^\tau}{\partial t} + \gamma(u_N^\tau - u_0^\tau) = \int_0^N f(u_t^\tau)dt.$$

Define $e(u_t^\tau) = \int_0^T g(u_t^\tau)d\tau - \delta(u_t^T - u_t^0)$, and $h(u_t^\tau) = \int_0^N f(u_t^\tau)dt - \gamma(u_N^\tau - u_0^\tau)$, we have:

$$\frac{\partial u_t^\tau}{\partial \tau} = e(u_t^\tau), \quad \frac{\partial u_t^\tau}{\partial t} = h(u_t^\tau),$$

thus,

$$u_t^\tau = \int_0^N h\left(\int_0^T e(u_t^\tau)d\tau\right)dt = \int_0^T e\left(\int_0^N h(u_t^\tau)dt\right)d\tau,$$

where $\frac{\partial e(u_t^\tau)}{\partial \tau} = g(u_t^\tau)$ and $\frac{\partial h(u_t^\tau)}{\partial t} = f(u_t^\tau)$.

## D  PROOF OF PROPOSITION 4

**Proposition 4** *Let $\mathbf{X}^n$ and $\mathbf{Y}^n$ be the node features, generated by Eq. 3, and $\Delta t \ll 1$. The gradients of the second-order Graph ODE $\mathbf{W}_l$ and second-order SNNs $\mathbf{W}^k$ are bounded as follows:*

$$\left|\frac{\partial \mathcal{L}}{\partial \mathbf{W}_l}\right| \leq \frac{\beta'\hat{\mathbf{D}}\Delta t(1 + \Gamma N \Delta t)}{v}\left(\max_{1 \leq i \leq v}(|\mathbf{X}_i^0| + |\mathbf{Y}_i^0|)\right)$$

$$+ \frac{\beta'\hat{\mathbf{D}}\Delta t(1 + \Gamma N \Delta t)}{v}\left(\max_{1 \leq i \leq v}|\bar{\mathbf{X}}_i| + \beta\sqrt{N\Delta t}\right)^2,$$

$$\left|\frac{\partial \mathcal{L}}{\partial \mathbf{W}^k}\right| \leq \frac{(1 + N\Gamma\Delta t)(1 + L\Theta\Delta\tau)V_{th}}{v\beta^2\Delta\tau}\left(\max_{1 \leq i \leq v}|\mathbf{X}_i^N| + \max_{1 \leq i \leq v}|\bar{\mathbf{X}}_i|\right).$$

*where $\beta = \max_x |\sigma(x)|$, $\beta' = \max_x |\sigma'(x)|$, $\hat{D} = \max_{i,j \in \mathcal{V}} \frac{1}{\sqrt{d_i d_j}}$, and $\Gamma := 6 + 4\beta'\hat{D}\max_{1 \leq n \leq T}||\mathbf{W}^n||_1$, $\Theta := 6 + 4\beta'\hat{D}\max_{1 \leq n \leq N}||\mathbf{W}^n||_1$. $d_i$ is the degree of node $i$, $\bar{\mathbf{X}}_i$ is the label of node $i$.*

*Proof.* Eq. 13 can be obtained from Rusch et al. (2022) directly. Then,

$$\frac{\partial \mathcal{L}}{\partial \mathbf{W}^k} = \frac{\partial \mathcal{L}}{\partial \mathbf{Z}_L^T}\frac{\partial \mathbf{Z}_L^T}{\partial \mathbf{Z}_l^T}\frac{\partial \mathbf{Z}_l^T}{\partial \mathbf{W}^k} = \frac{\partial \mathcal{L}}{\partial \mathbf{Z}_L^T}\prod_{n=l+1}^{L}\frac{\partial \mathbf{Z}_n^T}{\partial \mathbf{Z}_{n-1}^T}\frac{\partial \mathbf{Z}_l^T}{\partial \mathbf{W}^k}$$

$$= \frac{\partial \mathcal{L}}{\partial \mathbf{Z}_L^T}\prod_{n=l+1}^{L}\frac{\partial \mathbf{Z}_n^T}{\partial \mathbf{Z}_{n-1}^T}\frac{\partial \mathbf{Z}_l^T}{\partial \mathbf{Z}_l^k}\frac{\partial \mathbf{Z}_l^k}{\partial \mathbf{W}^k}$$

$$= \frac{\partial \mathcal{L}}{\partial \mathbf{Z}_L^T}\prod_{n=l+1}^{L}\frac{\partial \mathbf{Z}_n^T}{\partial \mathbf{Z}_{n-1}^T}\prod_{i=k+1}^{T}\frac{\partial \mathbf{Z}_l^i}{\partial \mathbf{Z}_l^{i-1}}\frac{\partial \mathbf{Z}_l^k}{\partial \mathbf{W}^k},$$

From Rusch et al. (2022), we have:

$$\left\| \frac{\partial \mathcal{L}}{\partial \boldsymbol{Z}_L^T} \right\|_\infty \le \frac{1}{v} \left( \max_{1 \le i \le v} |\boldsymbol{X}_i^T| + \max_{1 \le i \le v} |\bar{\boldsymbol{X}}_i| \right), \quad \left\| \frac{\partial \boldsymbol{Z}_L^T}{\partial \boldsymbol{Z}_t^T} \right\|_\infty \le 1 + L\Gamma\Delta t. \tag{23}$$

Due to the second-order SNN has a similar formulation to second-order GNN, we have a similar conclusion,

$$\left\| \frac{\partial \boldsymbol{Z}_l^T}{\partial \boldsymbol{Z}_l^k} \right\|_\infty \le 1 + T\Theta\Delta\tau, \tag{24}$$

with $\beta = \max_x |\sigma(x)|$, $\beta' = \max_x |\sigma'(x)|$, $\hat{D} = \max_{i,j \in \mathcal{V}} \frac{1}{\sqrt{d_i d_j}}$, and $\Theta := 6 + 4\beta' \hat{D} \max_{1 \le n \le N} ||\boldsymbol{W}^n||_1$, then:

$$\frac{\partial \boldsymbol{Z}_l^k}{\partial \boldsymbol{W}^k} \approx r(\boldsymbol{Z}_l^{k-1}) \le \frac{V_{th}}{\beta^2 \Delta\tau}, \tag{25}$$

where $r(\cdot)$ the spike representation operator defined in Eq. 12.

Multipling 23, 24 and 25, we have the upper bound:

$$\frac{\partial \mathcal{L}}{\partial \boldsymbol{W}^k} \le \frac{(1 + L\Gamma\Delta t)(1 + T\Theta\Delta\tau)V_{th}}{v\beta^2\Delta\tau} \left( \max_{1 \le i \le v} |\boldsymbol{X}_i^N| + \max_{1 \le i \le v} |\bar{\boldsymbol{X}}_i| \right). \tag{26}$$

# E  ABLATION STUDY

Table 4: Ablation results. **Bold** numbers mean the best performance.

| *Homophily level* | Cora 0.81 | Citeseer 0.74 | Pubmed 0.80 | Texax 0.11 | Wisconsin 0.21 | Cornell 0.3 | Avg. |
|---|---|---|---|---|---|---|---|
| CSGO-1st-2nd | 83.2±1.4 | 74.1±1.4 | 76.3±2.2 | 81.7±3.9 | 85.1±2.8 | 81.0±1.9 | 80.2 |
| CSGO-2nd-1st | 83.5±1.8 | 73.4±2.1 | 77.2±2.3 | 83.1±3.8 | 84.4±2.2 | 81.2±2.7 | 80.5 |
| CSGO-1st | 83.3±2.1 | 73.7±2.0 | 76.9±2.7 | 81.6±6.2 | 84.9±3.2 | 80.4±1.9 | 80.1 |
| CSGO-2nd | **83.7±1.3** | **75.2±2.0** | **79.6±2.3** | **87.3±4.2** | **88.8±2.5** | **83.7±2.7** | **83.1** |

We conducted ablation studies to assess the contributions of different components using two variants, and the results are presented in Table 4. Specifically, we introduced two model variants: (1) CSGO-1st-2nd, which utilizes the first-order SNNs and second-order Graph ODE, and (2) CSGO-2nd-1st, incorporating the second-order SNNs and first-order Graph ODE. Table 4 shows that (1) CSGO-2nd consistently outperforms other variations, while CSGO-1st-2nd yields the worst performance. This is because the issue of information loss is crucial for graph representation, and the incorporation of high-order SNNs assists in preserving more information, consequently achieving superior results. (2) In most cases, CSGO-2nd-1st outperforms both CSGO-1st and CSGO-1st-2nd, suggesting that, compared to the capability of Graph ODE in capturing dynamic node relationships, the ability to mitigate the issue of information loss is more important.

# F  SENSITIVITY ANALYSIS

In this part, we examine the sensitivity of the proposed CSGO to its hyperparameters, specifically the time latency parameter ($T$) in SNNs, which plays a crucial role in the model's performance. $T$ controls the number of SNNs propagation steps and is directly related to the training complexity. Figure 4 shows the results of $T$ across different datasets. We initially vary the parameter $T$ within the range of $\{5, 6, 7, 8, 9, 10, 11\}$ while keeping other parameters fixed. From the results, we find that, the performance exhibits a increasing trend initially, followed by stabilization as the value of $T$ increases. Typically, in SNNs, spiking signals are integrated with historical information at each time latency. Smaller values of $T$ result in less information available for graph representation, degrading the performance. However, large values of $N$ increase model complexity during training. Striking a balance between model performance and complexity, we set $T$ to 8 as default.

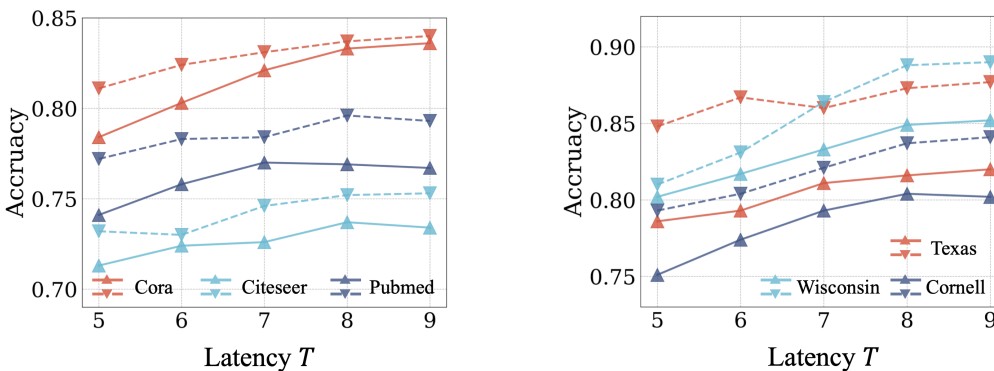

Figure 4: Sensitivity analysis on time latency $T$ in SNNs across various datasets. The solid line denotes the results of CSGO-1st, and the dotted line denotes the CSGO-2nd.

## G IMPACT STATEMENTS

This work introduces an innovative approach for continuous spiking graph neural network, with the objective of advancing the machine learning field, particularly in the domain of graph neural networks. The proposed method has the potential to substantially enhance the efficiency and scalability of graph learning tasks. The societal implications of this research are multifaceted. The introduced method has the capacity to contribute to the development of more efficient and effective machine learning systems, with potential applications across various domains, including healthcare, education, and technology. Such advancements could lead to improved services and products, ultimately benefiting society as a whole.

