# OpenReview forum: "Continuous Spiking Graph ODE Networks"
_ICLR.cc/2025/Conference — ICLR 2025 Conference Withdrawn Submission_

### Official Review · Reviewer_Y4Ym · 2024-10-28

**Soundness:** 2
**Presentation:** 2
**Contribution:** 2
**Rating:** 3
**Confidence:** 2

**Summary:**

The authors propose a new model that aims to combine Spiking Graph Neural networks (SGNNs) with ODE Networks. The goal is to extend SGNNs to dynamic graphs while maintaining low energy consumption. Furthermore they aim to fix the problem of information loss (occurring from binary sampling of features).

**Strengths:**

The methods energy efficiency is tested using a metric that compared traditional GPUs and neuromorphic chips and shows that the proposed methods are more efficient on neuromorphic chips than non spiking GNNs on raditional hardware.

**Weaknesses:**

1) The manuscript contains several ambiguities, making it hard to access the correctness, novelty and reproducibility. In detail:

a)  Motivation:
 -  Choice of Graph ODE and Higher-Order GNNs not well motivated / explained (see Introduction, paragraph3)

b) Clarity:
- It s not clear what a dynamic prediction task is (the methods imply a time dependent graph, but it is not clear what is time dependent)
- Problem Formulation in 3.1 is a general graph learning problem and has nothing to do with dynamic graphs.
- the terms SNN and SGN are used interchangeably and its unclear if it refers to the same method
- considering the last bullet point it is unclear, whether SNN/SGNN is an actural GNN, eq 4 seems to show that all neurons influence all other neurons, so where is the graph structure used
- “Higher-order” neural networks seems to be referencing to higher-order graph ODEs, which can be confusing to some readers who will first think of higher-order GNNs (e.g., Morris et al. 2019)
- What is the thought / motivation behind proposition 1? (unclear where eq comes from)
- How can one transform the unlabeled eq between 8 and 9 from eq 5?
- V_{th} It is not defined anywhere.

c) Experiments
- There are some inconsistencies with the results reported in the original papers for GCN and GAT (same data sets used)
- the data sets used in the experiments include static graphs only although the methods claims to handle dynamic graphs

**Questions:**

Dear authors,
after reading I believe the novelty of your research lies in the energy efficiency of you proposed method on neuromorphic chips, compared to traditional methods on GPU. Due to the many ambiguities, it is hard to access. Please explain more clearly what is the motivation behind your work? What is the addressed learning problem and why did you chose those data sets? Also, it would be helpful to add descriptive text to you equations, since not all Graph Learning/GNN researchers are familiar with spiking neural networks.

---

### Official Review · Reviewer_efTR · 2024-10-31

**Soundness:** 2
**Presentation:** 2
**Contribution:** 2
**Rating:** 5
**Confidence:** 3

**Summary:**

The paper presents Continuous Spiking Graph ODE Networks (CSGO), a novel approach combining SGNs with Graph ODEs to address energy efficiency and information loss in dynamic graph learning. However, several issues limit the paper’s clarity, applicability, and overall impact.

**Strengths:**

The paper proposes an approach that combines biologically-inspired SGNs with continuous Graph ODEs, a potentially impactful integration for energy-efficient applications requiring continuous graph evolution modeling.

**Weaknesses:**

1. Limited Novelty. The benefits of second-order ODEs in retaining information and addressing gradient stability are well-established in existing works.
2. Overstated Claims on Dynamic Graph Learning. The paper claims to address the problem of dynamic graph learning by combining SGNs with Graph ODEs, yet the experiments are conducted solely on static graph datasets. This discrepancy raises questions about the motivation for incorporating Graph ODEs. Besides, the comparison between the proposed method and traditional static SGNs demonstrates non-significant improvements, undermining the primary contribution of integrating Graph ODEs into SGNs.
3. Limited Comparison. For node classification tasks, this paper only chooses static node classification datasets. For graph classification tasks, this paper only evaluates on the MNIST dataset, which is not a standard benchmark for graph classification tasks.

**Questions:**

Please see Weaknesses.

---

### Official Review · Reviewer_5qr1 · 2024-11-03

**Soundness:** 2
**Presentation:** 1
**Contribution:** 3
**Rating:** 5
**Confidence:** 4

**Summary:**

The paper introduces a novel approach by integrating Spiking Graph Networks (SGNs) with Graph ODEs to create the CSGO framework, aimed at improving energy efficiency in graph learning tasks. The key contribution lies in leveraging spiking dynamics for continuous graph learning, which is particularly relevant for applications in low-power environments like wearable devices. The paper also provides a theoretical analysis of model stability, including gradient bounds. However, the specific real-world advantages, computational efficiency details, and evaluation in dynamic graph contexts need further elaboration to strengthen the contribution.

**Strengths:**

The paper presents a novel approach by integrating Spiking Graph Networks with Graph ODEs, which has the potential to bridge the gap between energy-efficient spiking mechanisms and continuous graph learning.

The emphasis on energy-efficient models, particularly in the context of spiking networks, is an important contribution for applications in low-power environments such as wearable devices and edge computing.

The paper provides theoretical analysis to support the stability of the proposed CSGO model, including bounds on gradients, which is an important step towards understanding the behavior of the integrated model.

**Weaknesses:**

The integration of Spiking Graph Networks (SGNs) with Graph ODEs is intriguing, but the overall contribution is vague. The scenarios where CSGO-1st and CSGO-2nd have advantages over traditional GNNs are not clearly articulated.

The results show that both CSGO-1st and CSGO-2nd outperform the CGNN baseline, which is unexpected since spiking networks typically struggle to surpass deep networks. There is no detailed explanation for this, and it's unclear if the CGNN baseline was optimized correctly. Also, the structure and interaction between SNN and Graph ODE components are not clearly explained. It's unclear how spike representations are generated at each step and whether the SNN and CGNN are compatible in terms of energy efficiency. The architecture appears fragmented.

The differentiable mapping in Equation (11) seems incorrect, particularly in its connection with Proposition 4. There is no justification for why these changes are valid, raising concerns about the mathematical correctness.

Despite the paper's focus on dynamic graph learning, the experiments primarily use static graph datasets (e.g., Cora, Citeseer, Pubmed). Without dynamic graphs, it's hard to validate the model's effectiveness in dynamic scenarios.

**Questions:**

1. Can you provide detailed calculations for Synaptic Operations (SOP) and MAC operations in SNNs, and clarify how the computational load compares to DNNs? How does your approach achieve efficiency despite time scale factors?

2. Could you explain the specific structure of the SNN module in more detail, including the inputs and how it interacts with Graph ODE? Are the neurons and their connections related to the parameters in the CGNN module, and if so, how?

3. Could you provide a more detailed explanation of how bounded gradients address vanishing gradient problems, particularly in comparison to previous work?

---

### Official Review · Reviewer_GbkR · 2024-11-12

**Soundness:** 3
**Presentation:** 3
**Contribution:** 2
**Rating:** 5
**Confidence:** 3

**Summary:**

The paper introduces Continuous Spiking Graph ODE Networks (CSGO), integrating Spiking Neural Networks (SNNs) with Graph Ordinary Differential Equations (Graph ODEs) for dynamic graph learning.

**Strengths:**

The model, particularly CSGO-2, addresses information loss through higher-order spike representations and differentiation.
Experiments on Cora, Citeseer, and Pubmed demonstrate competitive performance and energy efficiency.

**Weaknesses:**

CSGO assumes that the spike dynamics function g(uτ,τ) and Graph ODE function f(uτ,n) are continuous and differentiable. However, real-world data may contain discontinuities, potentially violating these assumptions and compromising model robustness. It would be advantageous for the authors to consider modifications that support piecewise continuous functions. Additionally, outlining specific strategies or algorithms to handle such discontinuities would significantly improve the model's resilience against imperfect data.

CSGO-2 relies on very small time steps (Δτ≪{τsyn,τmem}) to accurately capture spiking behaviour. This requirement may lead to increased computational demands, limiting the framework's scalability and feasibility in resource-constrained environments, and ultimately affecting energy efficiency. Addressing this issue by exploring optimization techniques could help balance the trade-off between accuracy and computational efficiency.

Evaluation could be strengthened by testing CSGO on more diverse datasets such as OGBN-Arxiv. Overall, CSGO presents a promising approach by merging SNNs with Graph ODEs for dynamic graph learning, addressing its assumptions on data continuity and computational scalability, as well as expanding its evaluation, is necessary to enhance its usability.

**Questions:**

1. How CSGO can deal with discontinuity in the real-world data?
2. What is the computational demand of relying on very small time-steps? What are the trade-offs between computational complexity and accuracy in terms time-step sizes?

---

### Note · Authors · 2024-12-03

I have read and agree with the venue's withdrawal policy on behalf of myself and my co-authors.